# Characterisation of the Effect of the Spatial Organisation of Hemicellulases on the Hydrolysis of Plant Biomass Polymer

**DOI:** 10.3390/ijms21124360

**Published:** 2020-06-19

**Authors:** Thomas Enjalbert, Marion De La Mare, Pierre Roblin, Louise Badruna, Thierry Vernet, Claire Dumon, Cédric Y. Montanier

**Affiliations:** 1Toulouse Biotechnology Institute (TBI), Université de Toulouse, CNRS, INRAE, INSA, 31077 Toulouse, France; enjalber@insa-toulouse.fr (T.E.); louise.badruna@hotmail.fr (L.B.); cdumon@insa-toulouse.fr (C.D.); 2Toulouse White Biotechnology, UMS INRA 1337, UMS CNRS 3582, Institut National des Sciences Appliquées de Toulouse, 31077 Toulouse, France; mdelamare@enobraq.com; 3Laboratoire de Génie Chimique, Université de Toulouse, CNRS, INPT, UPS, 31077 Toulouse, France; roblin@chimie.ups-tlse.fr; 4Institut de Biologie Structurale, Univ., Grenoble Alpes, CEA, CNRS, IBS, F-38000 Grenoble, France; thierry.vernet@ibs.fr

**Keywords:** xylanase, xylosidase, Bio Molecular Welding, spatial proximity, enzyme engineering, synergism

## Abstract

Synergism between enzymes is of crucial importance in cell metabolism. This synergism occurs often through a spatial organisation favouring proximity and substrate channelling. In this context, we developed a strategy for evaluating the impact of the geometry between two enzymes involved in nature in the recycling of the carbon derived from plant cell wall polymers. By using an innovative covalent association process using two protein fragments, Jo and In, we produced two bi-modular chimeric complexes connecting a xylanase and a xylosidase, involved in the deconstruction of xylose-based plant cell wall polymer. We first show that the intrinsic activity of the individual enzymes was preserved. Small Angle X-rays Scattering (SAXS) analysis of the complexes highlighted two different spatial organisations in solution, affecting both the distance between the enzymes (53 Å and 28 Å) and the distance between the catalytic pockets (94 Å and 75 Å). Reducing sugar and HPAEC-PAD analysis revealed different behaviour regarding the hydrolysis of Beechwood xylan. After 24 h of hydrolysis, one complex was able to release a higher amount of reducing sugar compare to the free enzymes (i.e., 15,640 and 14,549 µM of equivalent xylose, respectively). However, more interestingly, the two complexes were able to release variable percentages of xylooligosaccharides compared to the free enzymes. The structure of the complexes revealed some putative steric hindrance, which impacted both enzymatic efficiency and the product profile. This report shows that controlling the spatial geometry between two enzymes would help to better investigate synergism effect within complex multi-enzymatic machinery and control the final product.

## 1. Introduction

Enzymes are biological catalysts that increase the rate of chemical reactions within cells without being consumed or permanently altered by the reaction [1]. They are involved in a large number of important functions such as metabolic pathways in cells. In many cases, they are part of complex organisations that involve several enzymes with distinct catalytic activities, embedded in a cascade of reactions leading, for instance in the case of linear cascades, to the conversion of one substrate to one product [2,3,4]. However, a proper spatial organisation is required in order to provide suitable enzyme distance and active site orientation [5]. In nature, such parameters are encountered by (i) colocalising enzymes inside dedicated bacterial microcompartments [6], (ii) clustering enzymes in noncovalent dynamic complexes described as metabolon [7], and (iii) grafting enzymes on self-assembling scaffolding-based protein such as cellulosome [8].

Such protein organisations provide a beneficial effect to substrate channelling, mainly consisting of the transfer of reaction intermediates directly from one enzyme to another, thus decreasing the loss of reactants and intermediates with the bulk aqueous solvent [9]. However, recent studies suggest that the specific microenvironment created by enzyme proximity is a major determinant [10]. In the particular case of the plant cell wall carbohydrate-degrading cellulosome, the effect of the multienzyme cascade is slightly different, resulting also in an increase of synergistic effect rather than proper channelling [11]. Plant cell wall is a complex matrix of carbohydrate-based polymers such as cellulose, hemicelluloses, and pectins embedded with phenol-based lignin. The enzymes accomplishing the deconstruction of this matrix into single units of sugar are glycoside hydrolases (GHs), pectate lyases, carbohydrate esterases and other auxiliary enzymes [12,13]. The first description of the cellulosome, produced by the anaerobic bacterium *Clostridium thermocellum*, described an extracellular nanomachine of 2 MDa [14]. It is organised around a macro-molecular scaffold protein residing at the bacterial outer cell membrane and contains a Carbohydrate-Binding Module which binds to crystalline cellulose [15]. Nine receptor domains called the cohesins, connected by flexible linkers, interact with sub-nanomolar affinity with a complementary module fused to the GH called the dockerin [16]. The cellulosome composition is highly variable [17,18] and could also be organised in polycellulosomes at the surface of bacteria, hence displaying more than 100 catalytic domains. Previous work has also evidenced that the position of each GH docked to the scaffoldin is not random [19,20] and that spatial proximity is a key to the remarkable efficiency of the cellulosome [21,22]. The length itself of the linkers promotes spatial proximity and provides adaptation to the topology of complex polymers [23,24], bringing catalysts with complementary activity at specific site on the plant cell wall. Thus, cellulosomes are inherently dynamic structures in constant adjustment in order to adapt to the continuous degradation within the plant cell wall [25]. Therefore, it is challenging to assess the question of enzyme proximity at work in such context. Numerous publications investigated the spatial organisation in multi-enzyme cascades [26,27] and in our particular field involving GHs [28,29,30,31,32]. Nevertheless, the impact of the synergy on the chemical nature of the product [33,34] was rarely investigated, and to our knowledge, the spatial geometry between two GHs was never considered.

To shed light on the impact of the spatial organisation on enzyme synergy, we developed a new strategy to investigate the effect of the geometry between two GHs spatially close to each other. For this purpose, we used two engineered protein fragments, Jo and In, that spontaneously and covalently attach to form a complex [35]. The model enzymes of this study are two well-characterised GHs, the endo-β-1,4-xylanase GH11A from *Neocallimastix patriciarum* (*Np*Xyn11A) [35] and the β-1,4-xylosidase GH43 from *Bacillus halodurans* (*Bh*Xyl43) [36,37]. While *Np*Xyn11A released oligosaccharides from xylan, a linear polysaccharide consisting of β-1,4-linked xylose units with a large variety of side-chain substituents, *Bh*Xyl43 further hydrolyses the short oligomers of β-d-xylopyranosyl units that accumulate as a result of xylanase action. As Jo and In are covalently bound in a head to tail manner, by playing with the N- or C-terminal fusion of our enzymes, we produced chimeric bi-modular enzymes with two distinct spatial geometry, as revealed by SAXS analysis. Biochemical characterisation of the complexes on chromogenic substrates and polysaccharide from plants showed that subtle changes in the structure of the chimeric complex improve enzymatic efficiency and induce changes in product profiles compared to the free enzymes mixture in solution.

## 2. Results and Discussion

### 2.1. Production of the Recombinant Xylanase-Xylosidase Bi-Modular Complexes

The goal of this study was to engineer enzymes in a simple way in order to study the impact of the modulation of both geometry and spatial proximity between two catalysts to investigate their impact on synergism. For this purpose, the Bio Molecular Welding tool was used [35]. It consists of two small proteins Jo and In of 10 and 16 kDa respectively and able to bind to each other through a spontaneous and specific irreversible isopeptide bond involving the Lys191 from Jo and the Asn695 from In. The structure of this complex solved by crystallography (PDB: 5MKC) revealed an antiparallel organisation of about 5.5 nm long (Figure 1A). Hence, we made the assumption that expressing two enzymes of interest at both N-terminal of Jo and In or at the N-terminal of In and at C-terminal of Jo would contribute to modulate the orientation between the two proteins. Two well-characterised enzymes with complementary activity over xylan hydrolysis were elected: the endo-β-1,4-xylanase GH11A from *Neocalimastix patriciarum* (*Np*Xyn11A) [38] and the β-1,4-xylosidase GH43 from *Bacillus halodurans* (*Bh*Xyl43) [37]. The xylanase *Np*Xyn11A was expressed as a polyhistidine-tagged (His_6_-Tag) protein with or without In at its N-terminal leading to previously described _His_-In-*Np*Xyn11A or _His_-*Np*Xyn11A, respectively [34] (Figure 1B). The xylosidase *Bh*Xyl43 was expressed without any His_6_-Tag, in fusion with Jo at its N- or C-terminal, leading to soluble Jo-*Bh*Xyl43 and *Bh*Xyl43-Jo protein, respectively. This strategy was developed to allow in vitro formation following mixing of cell free extracts of complexes which could be recovered from the supernatant using the His_6_-Tag displayed only by the xylanase. Although a molar ratio of 1:1 between Jo and In is efficient enough to lead to up to 95% of the covalent complex Jo-In within 1 h [34], the formation of complex between _His_-In-*Np*Xyn11A and Jo-*Bh*Xyl43 or *Bh*Xyl43-Jo produced in the cell free extract was obtained with an optimised ratio of 1:5 as analysed on SDS-PAGE. This ratio can be explained by the possibility that the environment of few loops of Jo and In isolated from the originated multi-modular protein may be modified [35]. Actually, the intimate recognition process leading to the formation of the covalent bond involves flexible loops of Jo and In (N. Cox et al., manuscript in preparation) that participate to the formation of a hydrophobic pocket and thus promote a direct attack mechanism for isopeptide bond formation [39], in our case Lys191 of Jo and Asp600 and Asn695 for In [35]. A significant percentage of overexpressed chimeric proteins may display some distorted sub-structures which could explain the higher molar ratio of Jo containing enzymes, compare to _His_-In-*Np*Xyn11A. Nevertheless, resulting complexes _His_-In-*Np*Xyn11A-Jo-*Bh*Xyl43 and _His_-In-*Np*Xyn11A-*Bh*Xyl43-Jo (Figure 1B) were purified by IMAC followed by size exclusion chromatography, leading to pure complexes close to the theoretical molecular weight of 111.99 kDa and 111.86 kDa for _His_-In-*Np*Xyn11A-Jo-*Bh*Xyl43 and _His_-In-*Np*Xyn11A-*Bh*Xyl43-Jo respectively (Figure 1C). Enzymes were pure at >95% as evaluated by SDS-PAGE. 

### 2.2. Enzymatic Properties of the Xylanase and the Xylosidase Fused to Jo or In

Previous work has already demonstrated that introduction of In module at the N-terminal of *Np*Xyn11A did not affect its enzymatic activity [34]. To evaluate the impact of Jo to the activity of _His_-*Bh*Xyl43, an alternate of Jo-*Bh*Xyl43 was also expressed and purified with a His_6_-Tag at its N-terminal (_His_-Jo-*Bh*Xyl43). The optimal pH for _His_-Jo-*Bh*Xyl43 was determined at 8 using 4-nitrophenyl-β-d-xylopyranoside (*p*NP-X), which is in agreement with the optimal pH published for _His_-*Bh*Xyl43 under the same experimental conditions [37]. As the pH-optimum of enzymatic activity and stability are correlated [40], this measurement is a good indicator of a non-deleterious effect of the fusion of Jo to the xylosidase. Both xylanase and xylosidase optimum pH curves displayed a characteristic bell curve shape. The xylanase _His_-In-*Np*Xyn11A displayed an optimum pH = 6. Its residual activity at neutral pH was still consistent (68%), but was almost negligible at pH 8 (17%). However, the xylosidase displayed a high residual activity at pH 7 (92%) and to a lesser extent at pH 6 (66%) (Appendix A). Although previous work [34] showed that In had no effect on the specific activity of _His_-In-*Np*Xyn11A on 4-nitrophenyl-β-d-xylotrioside (*p*NP-X_3_), the effect of the fusion was evaluated on natural substrate in this study. *Np*Xyn11A kinetic parameters were determined using Beechwood xylan (BWX), a natural polymer of β-(1,4) xylose units partially substituted with charged 4-*O*-methyl glucuronic acid units (MeGlcA) (Table 1). The affinity of the wild type xylanase against BWX is 2.4 fold higher than _His_-In-*Np*Xyn11A, but the catalytic efficiency is very similar, with a *k*_cat_/*K*_M app_ value of 32.02 × 10^3^ and 25.60 × 10^3^ min^−1^·mg^−1^·mL, respectively. Indeed, like GH11 xylanase, *Np*Xyn11A is a globular protein of 24.7 kDa, displaying a large catalytic cleft within a β-jelly roll domain with connected loops in a right-hand shape partially closed [41]. The catalytic site is located in the palm, and the thumb is a flexible loop overlapping the catalytic site [41]. The In protein was fused to the N-terminal β-sheet constituting the outer part of the palm of the xylanase. The presence of In might narrow the space in the prolongation of the active site, and could slightly reduce the affinity for the substrate. However, we considered that the In module did not affect dramatically the activity of *Np*Xyn11A.

The kinetic parameters of _His_-Jo-*Bh*Xyl43 were determined using *p*NP-X. The affinity against the substrate is almost not affected by the presence of the additional Jo module compared to the wild type enzyme *Bh*Xyl43 (Table 1) [37]. The catalytic efficiency is comparable for both enzymes at 3014 and 2750 s^−1^·M^−1^, respectively. Thus, addition of the Jo domain fused at the N-terminal of *Bh*Xyl43 did not affect the catalytic parameters. We can reasonably assume that this is also the case for both Jo-*Bh*Xyl43 and *Bh*Xyl43-Jo. This is in accordance with the 3D structure of this 63 kDa enzyme (PDB:1YRZ, Fedorov et al., unpublished work), constituted by a five-bladed β-propeller fold for the catalytic domain, and an additional non-catalytic β-sandwich domain fused at its C-terminal. The active site possesses a pocket topology which is mainly constructed from residues of the β-propeller domain [42], located in the centre of the enzyme, without any obvious interaction with the N- or C-terminal of the protein.

### 2.3. SAXS Analysis Revealed Differences between the Structures of Each Complex in Solution

To investigate the spatial organisation of *Np*Xyn11A and *Bh*Xyl43 in the complexes _His_-In-*Np*Xyn11A-Jo-*Bh*Xyl43 and _His_-In-*Np*Xyn11A-*Bh*Xyl43-Jo, Small Angle X-rays Scattering (SAXS) was used to determine the structure of each chimeric complex in solution. Most of the GH43 catalytic domains are extended with a C-terminal domain which is required for the catalytic activity [42], and present a dimeric structure [42]. This is also the case for *Bh*Xyl43, as revealed by its 3D structure (PDB: 1YRZ). At first, SAXS data were collected from _His_-*Bh*Xyl43. As shown in Appendix A the SAXS data were fitted with the dimeric structure of the enzyme obtained by generating symmetry mates with Pymol software (Chi^2^ = 1.22), confirming the dimer form of the xylosidase in solution. 

Regarding the chimeric complexes, Figure 2 displays the superimposition of the SAXS curves. The most important differences are observed at intermediate angles (0.01–0.07 Å^−1^) and the calculation of the autocorrelation function clearly shows that the two constructs adopt different conformations in solution. According to biophysical parameters such as R_g_, D_max_, Porod’s volume and molecular weight (Appendix A, _His_-In-*Np*Xyn11A-Jo-*Bh*Xyl43 form appears more extended than the _His_-In-*Np*Xyn11A-*Bh*Xyl43-Jo form with bigger R_g_ and D_max_ values (R_g_: 56.4 > 50.5 Å and D_max_ 230 > 200 Å, respectively). On the other hand, the p(r) function of the _His_-In-*Np*Xyn11A-Jo-*Bh*Xyl43 form presents two distinct modulations correlated with the presence of two separated domains, not clearly visible on the p(r) of _His_-In-*Np*Xyn11A-*Bh*Xyl43-Jo form.

The Porod’s volume and the molecular weight are close for both complexes and the calculated molecular weights are consistent with a dimeric form of the chimeric enzymes, but the values are less than expected. Although an HPLC online with size exclusion chromatography was used to isolate the fraction corresponding to the peak, an equilibrium dimer-monomer is quickly established and the measured SAXS curves contains finally a fraction of monomeric form. As the dissociation phenomenon is observed for _His_-In-*Np*Xyn11A-Jo-*Bh*Xyl43 and _His_-In-*Np*Xyn11A-*Bh*Xyl43-Jo but not for the _His_-*Bh*Xyl43 dimer, we can suppose that the addition of new domains at the N- or C-terminal of the xylosidase could disturb slightly the association of the dimer.

As the structure of the domains present in both complexes are known, rigid body molecular modelling against SAXS data was performed using SASREFMX program [33] to propose models that could explain the differences observed in solution. In total, 10 runs were performed for each chimeric enzyme. For each run, a very good Chi^2^ value was obtained (2.1 and 1.7, for _His_-In-*Np*Xyn11A-Jo-*Bh*Xyl43 and _His_-In-*Np*Xyn11A-*Bh*Xyl43-Jo, respectively) and the three extreme conformations proposed by the program were superimposed to check the variability of the solution (Appendix A).

The proposed models structurally mimic the SAXS data very well (Figure 3). The structural differences between the two complexes illustrate the differences observed in the measured biophysical parameters and in the pair distribution function p(r) calculation. The complex _His_-In-*Np*Xyn11A-Jo-*Bh*Xyl43 presents an extended arc shaped structure with the catalytic pocket of the xylosidase in the middle, pointing outside the curvature, and the catalytic pocket of the xylanase facing one another at the extremity of the complex (Figure 4A). The chimer _His_-In-*Np*Xyn11A-*Bh*Xyl43-Jo displays a much more compact structure, as an “M” shape. The catalytic site of the xylosidase points out from the middle of the structure, whereas the catalytic site of the xylanase faces down in the opposite direction at the extremity (Figure 4B). 

This rigid body modelling approach allowed us to finally propose a low-resolution spatial organisation of the different catalytic domains and to measure the geometric influence of Jo fused at the N- or C-terminal *Bh*Xyl43. A closer view of the spatial organisation of each GH within a monomer is displayed in Figure 4. Each complex overlapped the Jo-In domain. In the complex _His_-In-*Np*Xyn11A-Jo-*Bh*Xyl43 (Figure 4C), the two catalytic pockets are distanced by ~94 Å and faced the same side of Jo-In, which distanced the two GHs by ~53 Å. On the other hand, the complex _His_-In-*Np*Xyn11A-*Bh*Xyl43-Jo (Figure 4D) presents the two GHs side by side, distanced by ~28 Å. Their respective catalytic pockets are one behind the other and separated by ~75 Å. The Jo-In association and its head to tail organisation therefore provide an original and efficient strategy for controlling the geometric organisation between two linked GH.

### 2.4. Impact of the Chimeric Enzyme Complexes on the Enzymatic Activity

Although the addition of In or Jo modules to the sequence of the xylanase and the xylosidase did not significantly affect their respective catalytic efficiency, we characterised the activity of each GH within their bi-modular association. For that purpose, the SA of the discrete xylanase and xylosidase was compared to each respective enzyme within the bi-modular complexes _His_-In-*Np*Xyn11A-Jo-*Bh*Xyl43 and _His_-In-*Np*Xyn11A-*Bh*Xyl43-Jo (Table 2). By definition, a specific activity is defined as the number of µmoles of product formed per minute and per milligram of protein (IU/mg). However, to properly compare the amount of catalytic site present during the reaction, a molar concentration of enzyme was considered instead of a mass concentration, resulting in a specific activity expressed as µmoles of product formed per minute and per µmoles of enzyme (IU/µmole). For clarity purposes, the term specific activity was conserved. Data demonstrated that the activity of *Np*Xyn11A was preserved when it was covalently bound to Jo-*Bh*Xyl43. Actually, the complex _His_-In-*Np*Xyn11A-Jo-*Bh*Xyl43 displays an increased xylanase activity (1404.7 IU/µmole) compared to _His_-In-*Np*Xyn11A (1251.9 IU/µmole). This is probably due to the lesser extent of xylosidase’s ability to recognise *p*NP-X_3_ as a substrate (~100-fold less active on *p*NP-X_3_ than *p*NP-X, Table 2). *p*NP-X_3_ and *p*NP-X are widely used to monitor xylanase and xylosidase activity, respectively. However, previous publications have established that *Bh*Xyl43 is active on xylobiose (X_2_), xylotriose (X_3_) and xylotetraose (X_4_) [36,37], but also on *p*NP-X_3_ and even xylohexaose (X_6_) (Appendix A respectively). The xylosidase is not active on XA^3^XX [37] and A^2^XX (Figure 4C), confirming its ability to release xylose only from the non-reducing end sugar (following the oligosaccharide nomenclature of Fauré et al. [43]). Although the SA of _His_-*Bh*Xyl43 was preserved when mixed with _His_-In-*Np*Xyn11A (222.9 IU/µmole compared to 211.6 IU/µmole), the xylosidase activity was reduced by 1.8 fold and 1.2 fold within the complexes _His_-In-*Np*Xyn11A-Jo-*Bh*Xyl43 and _His_-In-*Np*Xyn11A-*Bh*Xyl43-Jo, respectively, compared to the xylosidase alone. Considering the relatively small size of the *p*NP-X, this result is surprising, as the substrate should diffuse easily in solution. An explanation may be provided by the structure of the complexes in solution solved from the SAXS data (Figure 4). Unlike for _His_-In-*Np*Xyn11A-*Bh*Xyl43-Jo, the catalytic pocket of the xylosidase is buried in the middle of the complex _His_-In-*Np*Xyn11A-Jo-*Bh*Xyl43, where the branches of the arc constituted with Jo-In and *Np*Xyn11A may induced some unspecific interactions with the substrate, reducing the efficiency of the catalyst.

### 2.5. Characterisation of the Complexes on Plant Cell Wall Polysaccharide

*Np*Xyn11A and *Bh*Xyl43 displayed different pH optimum over aryl-β-xylosides, pH 6 and 8 respectively. Therefore, SA against BWX was determined at pH 6, 7 and 8 in order to compare the activity of the complexes to _His_-In-*Np*Xyn11A or _His_-*Bh*Xyl43 free in solution. To do so, we used an equimolar mixture of both GH and each bi-molecular enzymes _His_-In-*Np*Xyn11A-Jo-*Bh*Xyl43, and _His_-In-*Np*Xyn11A-*Bh*Xyl43-Jo (Table 3). The equivalent amount of reducing xylose units released from BWX was measured by dinitrosalicylic acid assay (DNSA) [44]. Experiments were conducted at 37 °C, the optimal temperature of *Np*Xyn11A [43], different to the optimal temperature of *Bh*Xyl43 (45 °C) [37] at which the xylanase is not stable. As expected, _His_-*Bh*Xyl43 was poorly active on BWX (0.36 × 10^3^ UI/µmole) and _His_-In-*Np*Xyn11A displayed the best activity at pH 6 (47.11 × 10^3^ UI/µmole). However, unlike on *p*NP-X_3_ (Appendix A), the xylanase activity on BWX remained high at pH 8 (39.98 × 10^3^ UI/µmole, 84% activity remaining). Adding 1% of BWX in the buffered solution did not affect the final pH. Therefore, the determination of the pH optimum for _His_-In-*Np*Xyn11A in presence of BWX was performed, revealing a better activity at pH 6 and 8, while activity at pH 7 was slightly diminished (Appendix A). Actually, the modification of the acid dissociation constant p*K*_a_ of the catalytic acid is a critical parameter for GH activity and the environment, such as mutation, changing in net charge of the catalytic residue or solvent accessibility could modulate the value of the p*K*_a_ and cause a shift toward higher pH [45,46,47,48,49]. BWX is much larger and complex compared to *p*NP-X_3_ and fit in both the glycone and aglycone part of the catalytic site, with two negatively charged MeGlcA side chains possibly accommodated by the −3 and/or +2 catalytic subsite [50,51], modifying the environment close to the active site [52]. The system of the buffer used might also modulate the value of the p*K*_a_ of the catalytic residues. Altogether, these results suggest that *Np*Xyn11A could be considered as an alkaline xylanase (Asn proximal to the catalytic Glu [52]) displaying a relatively large range of pH optimum activity as previously reported [53,54,55]. Compared to the activity of _His_-In-*Np*Xyn11A alone, the reaction mixture supplemented with _His_-*Bh*Xyl43 displayed a very similar profile of SA at all considered pH value, while the complex _His_-In-*Np*Xyn11A-Jo-*Bh*Xyl43 clearly displayed the lowest specific activity at all pH (around 33 × 10^3^ UI/µmole) (Table 3). _His_-In-*Np*Xyn11A-*Bh*Xyl43-Jo displayed the best overall specific activity at pH 7 (48.21 × 10^3^ UI/µmole). To evaluate the contribution of each enzymatic mixture, xylooligosaccharides released during the hydrolysis of BWX were monitored by HPAEC-PAD. It is clear that X_3_ is produced at the very first step of the reaction, while X_1_ started to be accumulated after 60 min of reaction (Appendix A). Accumulation of X_3_ by a xylan GH11 degrading enzyme is a marker of a xylanase activity [56], just like X_1_ is for GH43 [57]. This observation reflects the fact that enough small oligosaccharides have to be produced by the xylanase first before being hydrolysed by the xylosidase. As the SA were determined in the same experimental conditions, during the initial rate of the reaction (within the 40 first minutes), we can reasonably assume that the SA measured (Table 3) reflect mainly the xylanase activity. Thus, differences in SA between the two complexes may be explained by spatial organisation, more or less impeding the access of the substrate to the catalytic site of *Np*Xyn11A. As revealed by SAXS analysis, the structure of _His_-In-*Np*Xyn11A-*Bh*Xyl43-Jo is more compact (Figure 4B), with the xylanase domains distanced by 105 Å, whereas the structure of _His_-In-*Np*Xyn11A-Jo-*Bh*Xyl43 is more extended with the xylanase domains distanced by 175 Å. The closest proximity of the two xylanase domains may favour enzymatic hydrolysis events, resulting in a better SA.

### 2.6. Analyses of Xylan Degradation Products by the Chimeric Bi-Modular Enzymes

To more deeply investigate the effect of the different spatial organisations in the chimeric complexes, analysis of the final product released during 24 h of hydrolysis of BWX at different pH was performed. Products were analysed by HPAEC-PAD (Figure 5) and DNS assay (Figure 6). HPAEC-PAD made it possible to accurately quantify the release of soluble xylooligosaccharides ranging from xylose (X_1_) to xylohexaose (X_6_). Meanwhile, the colorimetric assay quantified in a non-discriminatory manner soluble xylan-reducing ends, including both xylose and xylooligosaccharides, but also longer oligomers with various amounts of substitution, as previously determined by MALDI-ToF mass spectrometry under similar conditions [34]. Both *Np*Xyn11A and *Bh*Xyl43 were stable, displaying a reduction of only 7% of their respective activity after 24 h (Appendix A). As expected, the activity of *Bh*Xyl43 against BWX was almost indistinguishable (Appendix A) and almost no X_1_ was produced by the activity of the xylanase alone (Figure 5). The contribution of the xylosidase to the xylanase activity was observed directly by the release of X_1_, at first modestly during the 60 first minutes then more obviously after 240 min (Figure 5). After 24 h at pH 7 (Figure 5B), X_1_ represents 1.6% of the total amount of xylooligosaccharides released from BWX by _His_-In-*Np*Xyn11A compared to 59% when the xylanase is supplemented with _His_-*Bh*Xyl43. A reduced amount of X_3_, combined with an increase of X_2_ was concomitant to the production of xylose by the *Bh*Xyl43. This accumulation of xylose revealed the specificity of the xylosidase, which is able to hydrolase small oligosaccharides produced by the xylanase [58]. Intermediate xylooligosaccharides ranging from X_4_ to X_6_ were not accumulated as they are substrate for both enzymes. The total amount of xylooligosaccharides reached its maximum at pH 7 after 24 h by the free xylanase and xylosidase mixture (9735 µM, Figure 5B). pH 8 had a dramatic effect on xylooligosaccharides production (6660 µM, Figure 5C), as did pH 6, although to a lesser extent (7213 µM, Figure 5A). This observation suggests that neutral pH preserved enough activity of both enzymes. Regarding the enzymatic activity of the complexes, both followed the same pH dependency as the mixture of the two GHs in solution. However, _His_-In-*Np*Xyn11A-*Bh*Xyl43-Jo released less total xylooligosaccharides compared to _His_-In-*Np*Xyn11A-Jo-*Bh*Xyl43 after 24 h, as exemplified at pH 7, with 8694 µM and 9016 µM, respectively, but was able to generate at least 50% more xylose, whatever the pH value (Figure 5B). On the contrary, the amount of X_3_ produced was higher for _His_-In-*Np*Xyn11A-Jo-*Bh*Xyl43 than for _His_-In-*Np*Xyn11A-*Bh*Xyl43-Jo (1428 µM and 3191 µM, respectively, at pH 7). 

Total reducing end sugars were measured by DNSA, resulting in a concentration of equivalent xylose units expressed in µM. From these values, Figure 6 represents the total amount of xylose units, including the concentration of X_1_ and the sum of xylooligosaccharides from X_2_ to X_6_ as measured by HPAEC-PAD. The difference consists of polymers with a degree of polymerisation (DP) > 6. Regarding the total amount of reducing end sugars after 24 h, the chimeric enzyme _His_-In-*Np*Xyn11A-Jo-*Bh*Xyl43 was the most productive compared to the other enzymes, whatever the pH. At neutral pH after 24 h, _His_-In-*Np*Xyn11A released 11,279 µM of equivalent xylose, compared to 14,549 µM when the reaction medium was supplemented with _His_-*Bh*Xyl43, 13,596 µM with _His_-In-*Np*Xyn11A-*Bh*Xyl43-Jo, and 15,640 µM with _His_-In-*Np*Xyn11A-Jo-*Bh*Xyl43 (Figure 6B). Considering the effect of the xylanase activity only with respect to the corresponding concentration of equivalent amount of reduced xylose of oligomers of DP > 6 (lower DP may be provided by xylosidase activity), the complex _His_-In-*Np*Xyn11A-Jo-*Bh*Xyl43 was the most active, with 6623 µM of equivalent xylose at pH 7, whereas _His_-In-*Np*Xyn11A released 5933 µM, _His_-In-*Np*Xyn11A + _His_-*Bh*Xyl43 4757 µM, and _His_-In-*Np*Xyn11A-*Bh*Xyl43-Jo 4858 µM.

Table 4 presents the percentage of xylooligomers as measured by HPAEC-PAD from the total amount of reducing sugar determined by DNSA, for the different enzymes at pH 7 after 24 h. Neither X_5_ nor X_6_ is accumulated under any of experimental conditions, reflecting the activity of both _His_-In-*Np*Xyn11A and _His_-*Bh*Xyl43 on short xylooligosaccharides. However, this table highlights differences in enzymatic mechanism. Actually, _His_-In-*Np*Xyn11A produced mostly X_3_ (29.8%), around 7% of X_2_ and X_4_ and no X_1_. When supplemented with the xylosidase, the proportion of xylooligosaccharides was very similar to those produced by the complex _His_-In-*Np*Xyn11A-*Bh*Xyl43-Jo. The main product was X_1_ (39.5% and 32.7%), to a lesser extent X_2_ (16.4% and 17.3%) and X_3_ (7.8% and 10.5%), and with only 3% of X_4_. On the contrary, _His_-In-*Np*Xyn11A-Jo-*Bh*Xyl43 presents a mid-way profile with an equivalent percentage of both X_1_ and X_3_ (15.8% and 20.4%) and a similar percentage of X_2_ and X_4_ (11.8% and 8.4%). 

## 3. Materials and Methods

### 3.1. Gene Cloning

The xylanase 11A from *Neocallimastix patriciarum* (*Np*Xyn11A) cloned into pET22b [38] was sub-cloned in pET28b in fusion with In as previously described [34]. The xylosidase 43 from *Bacillus halodurans* C-125 is cloned in pET28b (pET28b-BH3683, [37]). The resulting genes code for the C-terminal His-tagged _His_-*Np*Xyn11A, the N-terminal His-tagged fusion proteins _His_-In-*Np*Xyn11A and _His_-*Bh*Xyl43, respectively. The region of the *jo* gene was amplified by PCR from the plasmid pBMW2 [35] using the following primers: 5′ GGA GAT ATA CCA TGG GCA GCA GCC ATC ACC AT CATC 3′ and 5′ GAT TGA CCA TGC TAG CGC TGC CGC GCG GCA CCA GGT CGA CGT TAT CTT TTT GTT CAT ACA CTG TTT TCC C 3′, which introduce a NcoI and a NheI site. The PCR product was introduced by homologous recombination (In-Fusion^®^ HD cloning kit, Clonetech, Mountain View, CA, USA) into a pET28b-BH3683 linearized vector using NcoI and NheI to produce pET28-Jo-BhXyl43-His. The resulting gene codes for the N-terminal His-tagged protein _His_-Jo-*Bh*Xyl43. The genes coding for Jo-*Bh*Xyl43 and *Bh*Xyl43-Jo were synthetized and subcloned in a pET28a linearized vector using NcoI and HindIII by GenScript HK Limited (Piscataway, NJ, USA), resulting in protein without a polyhistidine tag (Appendix A). 

### 3.2. Production, Expression and Purification

To express *Np*Xyn11A-_His_, _His_-*Bh*Xyl43 and derivate enzymes, *Escherichia coli* strain BL21 (DE3) harbouring the respective plasmids was cultured to mid-exponential phase (*A*_600nm_ 0.6) in Luria-Bertani broth at 37 °C. Expression of the recombinant enzyme was induced by the addition of isopropyl-β-d-thiogalactopyranoside to a final concentration of 1 mM and further incubation for 4 h at 37 °C. Harvested cells (10 min, 5000× *g*) were resuspended in 50 mM phosphate buffer, pH 7.2 containing 300 mM NaCl and a protease inhibitor cocktail (SigmaFAST protease inhibitor cocktail, Sigma-Aldrich, Darmstadt, Germany). The cells were lysed by sonication on ice for 1 min. The lysate was clarified by centrifugation (30 min at 74,000× *g* at 4 °C). Proteins harbouring a His_6_-Tag were purified by immobilised metal ion affinity chromatography (IMAC) using Talon resin (TALON^®^ Metal Affinity Resin, Clontech) and elution in 50 mM sodium phosphate buffer, pH 7.2 containing 300 mM NaCl and 150 mM imidazole. The eluted proteins were extensively dialyzed against 50 mM sodium phosphate buffer, pH 7.2. Purified proteins were adjudged homogenous by SDS-PAGE (Any kD™ Mini-PROTEAN^®^ TGX Stain-Free™ Protein Gels, Bio-rad, Hercules, CA, USA). Protein concentrations were determined by measuring absorbance at 280 nm and applying the Lambert-Beer equation. Theoretical molar extinction coefficients and molecular weight were calculated using ProtParam online software (https://web.expasy.org/protparam/) [59] (Appendix A).

### 3.3. Covalent Association of _His_-In-NpXyn11A and BhXyl43

The relative amount of _His_-In-*Np*Xyn11A, Jo-*Bh*Xyl43 and *Bh*Xyl43-Jo present in cell free extracts (filtered at 0.45 µm) was evaluated by SDS-PAGE using Image Lab™ Software (version 6.0.1 build 34, Bio-rad). Usually, one volume of cell free extract of _His_-In-*Np*Xyn11A was mixed to five volume of cell free extract of Jo-*Bh*Xyl43 or *Bh*Xyl43-Jo and incubated at room temperature for 90 min under constant agitation. Complexes were removed from the solution by immobilised metal ion affinity chromatography as described above and eluted with increasing concentrations of imidazole (5 mM, 200 mM and 300 mM). Traces of non-complexed _His_-In-*Np*Xyn11A were removed by a final round of purification carried out using a XK16 HiLoad 16/600 Superdex HiLoad S200 prep-grade gel filtration column (GE Healthcare Life Sciences, Chicago, IL, USA) connected to an Äkta Pure system. Typically, 3 mL of protein were loaded onto the column at 1 mL/min and finally eluted from the column using 50 mM sodium phosphate buffer, pH 7 containing 150 mM NaCl. Purified proteins were adjudged homogenous by SDS-PAGE. 

### 3.4. Enzymatic Activity

The kinetics parameters *K*_M_, V_max_, *k*_ca_t, and *k*_cat_/*K*_M_ of _His_-Jo-*Bh*Xyl43 were calculated from initial velocities at substrate concentration of 4-nitrophenyl-β-d-xylopyranoside (*p*NP-X, Carbosynth Ltd., Compton, UK) varying from 0.1 to 18 mM. Assays were conducted in 50 mM Tris-HCl pH 8 supplemented with 1 mg/mL BSA, at 45 °C for 10 min as previously described (10.1007/s00253-006-0512-5). Specific activities of xylosidase (50 nM) were determined using 5 mM of *p*NP-X for 15 min at 37 °C, in 50 mM Tris-HCl pH 8 supplemented with 1 mg/mL BSA. Specific activities of xylanase (100 nM) were determined using 5 mM of 4-nitrophenyl-β-d-xylotrioside (*p*NP-X_3_, LLC “Institute of road surfaces”) for 15 min at 37 °C, in 50 mM phosphate pH 7.2 supplemented with 1 mg/mL BSA [38]. Absorbance at 401 nm of the released 4-nitrophenolate (ε = 12,578 M^−1^·cm^−1^) was measured in quartz cuvettes with a chamber volume of 500 μL (cuvettes Hellma Analytics), using a spectrophotometer Cary 100 Bio (Agilent Technology, Santa Clara, CA, USA). One unit of β-xylosidase activity or endo-xylanase activity was defined as the amount of enzyme releasing 1 μmol of *p*NP per minute using the defined conditions. To determine the optimum pH of _His_-Jo-*Bh*Xyl43, 5 mM of *p*NP-X was used and reactions were performed at 45 °C for 15 min under constant agitation at 1400 rpm (ThermoMixer^®^ C, Eppendorf, Hamburg, Germany) in 2 mL centrifuge tube, in presence of 1 mg/mL of BSA. An aliquot of 50 μL was withdrawn and instantly mixed with 200 μL of 1 M Na_2_CO_3_. To vary pH from 4 to 5, 50 mM citrate was used, from 6 to 7, 50 mM phosphate buffer was used, from pH 8 to 9, 50 mM bicine was used and at pH 10, glycine/NaOH buffer was used. Absorbance at 401 nm (ε = 22,209 M^−1^·cm^−1^) was measured using a microplate spectrophotometer (Eon Microplate Spectrophotometer, Biotek Instruments, Winooski, VT, USA).

Dinitrosalicylic acid assay (DNSA) was performed to estimate the concentration of reducing sugar equivalent to xylose when beechwood xylan (BWX, Megazyme, Bray, Ireland) was the substrate. BWX solution was prepared by mixing the dry powder with water during at least 30 min at 90 °C. The enzymatic reactions were performed at 37 °C under constant agitation at 1400 rpm (ThermoMixer^®^ C, Eppendorf) in 2 mL centrifuge tube. At the regular time, an aliquot of 100 μL was mixed with 100 μL of DNS and incubated for 10 min at 95 °C. After cooling down on ice, 1 mL of deionised water was added and absorbance at 540 nm was measured (microplate reader Tecan Infinite M200 PRO). The apparent kinetics parameters *K*_Mapp_, V_max_, *k*_ca_t, and *k*_cat_/*K*_Mapp_ of *Np*Xyn11A were calculated from initial velocities at substrate concentration of BWX varying from 0.3 to 15 mg/mL as previously described [38] in 12 mM sodium citrate, 50 mM sodium phosphate buffer pH 6 supplemented with 1 mg/mL of BSA. Specific activities were determined using BWX at 1% and 1 nM of enzymes in 50 mM phosphate buffer supplemented with 1 mg/mL BSA, during 1 h at 37 °C. The pH of the reaction mixtures was adjusted to 6, 7 or 8 after adding the substrate using orthophosphoric acid or NaOH. For the kinetics over 24 h, aliquots of 150 µL were pipetted out at regular times and inactivated by heating at 95 °C for 5 min. Samples were split in two and stored at −20 °C until further analysis by HPAEC-PAD (see below). A sample volume of 25 µL was added to 25 µL of DNS and incubated for 10 min at 95 °C. A volume of 250 µL of deionised water was added before absorbance at 540 nm was measured (microplate reader Tecan Infinite M200 PRO). A standard curve was prepared accordingly. All experiments were performed in triplicate, and the reported values are the means of three experiments. Mathematical calculations and kinetic parameters derived from Michaelis-Menten representations were performed using the software SigmaPlot 11.0 (Systat Software, San Jose, CA, USA)

### 3.5. HPAEC-PAD

Quantification of short xylooligosaccharides released over the time from BWX by complexes and free enzymes were determined using aliquots removed at regular time intervals and heated at 95 °C for 5 min to terminate the reaction. Each sample (5 to 50 µL, depending on the progress of the reaction) was centrifuged at 20,000× *g* for 5 min and quantified by HPAEC-PAD using a Dionex ICS 3000 dual chromatography system. Xylooligosaccharides were separated on a Carbo-Pac PA-100 guard and analytical column PA-100 (2 × 50 mm and 2 × 250 mm). Separation of oligosaccharides was achieved by isocratic elution with 100 mM NaOH at a flow rate of 1 mL/min from 0 to 10 min, a gradient of 0 to 75 mM sodium acetate in 100 mM NaOH from 10 min to 25 min, and isocratic elution with 500 mM sodium acetate in 100 mM NaOH from 25 min to 35 min, then the column was re-equilibrated with 100 mM NaOH for another 10 min. Calibration was achieved using xylose and xylooligosaccharides (X, X_2_, X_3_, X_4_, X_5_ and X_6_) at concentrations from 5 μg/mL to 40 μg/mL. All experiments were performed in triplicate, and reported values are the means of three experiments (Dionex™ Chromeleon™ 7.2, ThermoFisher, Waltham, MA, USA). 

### 3.6. Small Angle X-ray Scattering

Before analysis, samples were buffer exchanged using PD-10 column (GE Healthcare) in 50 mM phosphate pH 7. Proteins were concentrated to around 10 mg/mL using centrifugal filter devices (Amicon^®^ Ultra 30 or 50K, Merck KGaA, Darmstadt, Germany) and final concentration values were measured (NanoDrop™ 2000, ThermoFischer). SAXS measurements were performed at Laboratoire de Genie Chimique, Toulouse, on the XEUSS 2.0 bench with a copper internal source (Genix3D) that produces an X-ray beam with an energy of 8keV and a flow of 30.106 ph·s^−1^ with a beam size resolution close to 500 × 500 μm. Samples were pipetted (volume of 50 µL) from the sample holder maintained at a constant temperature using a circulating water bath, to the measurement cell placed under vacuum to limit air absorption. Alternatively, an HPLC with a size exclusion column online is coupled to the SAXS to remove the aggregate and obtain a SAXS curve from monodisperse solution. Data were collected on 150 × 150 mm area DECTRIS detector (Pilatus 1M) at a sample to detector distance of 1.216 m, giving a range comprised from 0.005 to 0.5 Å^−1^. Then each scattering curve obtained for every sample is an average of at least 6 measurements with a data collection time of 1800 s. The averaged curves obtained with direct injection and SEC-HPLC are merged to obtain a composite curve with no aggregation contribution at small angles and a low noise at high angles. Finally, to obtain the absolute scattering intensity I(q) for the solutes, the background with the buffer solution contribution was subtracted from the total SAXS profile. The data integration and reduction were performed with FOXTROT. The biophysical parameters such as gyration radius Rg, maximal distance Dmax and Porod volume were calculated by using PRIMUS [60] from ATSAS suite, and the rigid bodies molecular modelling against SAXS was performed with SASREF [61].

## 4. Conclusions

This work demonstrated the possibility of using the Jo-In modules to create chimeric enzyme with modulated spatial organisation as demonstrated by SAXS analysis. Introduction of Jo or In as fusion to the xylanase *Np*Xyn11A or the xylosidase *Bh*Xyl43 did not alter their respective kinetic parameters. However, these Jo and In modules were of major interest in covalently associating the enzymes and creating chimeric complexes, the structures of which were solved in solution by SAXS. The arc shape or the “M” shape of the complexes resulting in a different positioning of the Jo module fused to *Bh*Xyl43 impacted the activity of each chimeric enzyme. Of course, a xylanase supplemented with a xylosidase increases the total activity, as measured by total reducing end sugar compared to xylanase alone [58]. However, by covalently associating the two GHs, we were able to modulate the SA or to increase the amount of hydrolytic events, as measured by DNSA. The arc shape probably presents a more solvent-exposed catalytic pocket of the xylanase, whereas the “M” shape presents a more compact structure, with the catalytic pocket of the GH one behind the other. Beyond the kinetic parameters, the two chimeric structures display some differences in product profile. Indeed, _His_-In-*Np*Xyn11A-Jo-*Bh*Xyl43 presents the lowest SA on *p*NP-X and BWX. However, this complex generated the highest amount of total reducing end sugar, with a high percentage of X_1_ and X_3_. On the other hand, _His_-In-*Np*Xyn11A-*Bh*Xyl43-Jo maintained a relatively stable SA on *p*NP-X and BWX compared to the two GHs free in solution. However, after 24 h, its activity measured by DNSA on BWX was low, releasing an equivalent amount of X_1_ and X_3_ compared to *Np*Xyn11A and *Bh*Xyl43 in free solution. However, the structure of the two complexes revealed some putative steric hindrance that impacted both enzymatic efficiency and product profile, and which still needs to be elucidated. This intimate correlation between the enzymatic activity and the structure of the complexes in solution provides some evidence that controlling the distance, but also the spatial geometry, between two GHs is of importance in controlling enzymatic processes.

## Figures and Tables

**Figure 1 ijms-21-04360-f001:**
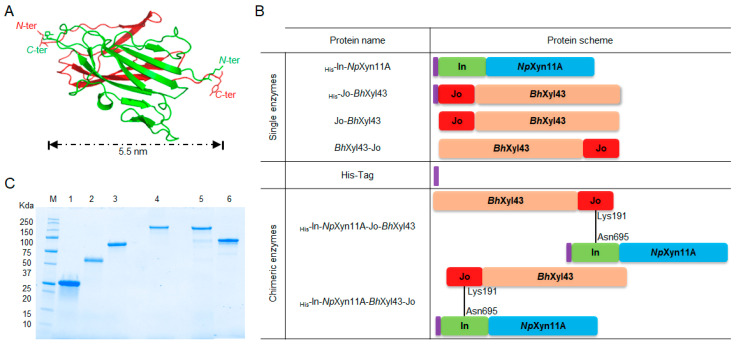
Expression and purification of the bi-modular enzymes. (**A**) Covalent association of Jo (red) and In (green) through the isopeptide bond of the Jo-Lys191 and In-Asn695 (black), displaying the head to tail orientation of the N- and C-termini (from PDB 5MKC). (**B**) Domain organisation of the Jo and In derivates of *Np*Xyn11A and *Bh*Xyl43, before and after the association through the Jo-In proteins. His-tag is in purple. (**C**) SDS-PAGE of the protein used in this study. Lanes: M, molecular mass markers; 1, _His_-*Np*Xyn11A; 2, _His_-In-*Np*Xyn11A; 3, _His_-*Bh*Xyl43; 4, _His_-In-*Np*Xyn11A-Jo-*Bh*Xyl43; 5, _His_-In-*Np*Xyn11A-*Bh*Xyl43-Jo; 6, _His_-Jo-*Bh*Xyl43 (image made from two SDS-PAGE, see Appendix A for details).

**Figure 2 ijms-21-04360-f002:**
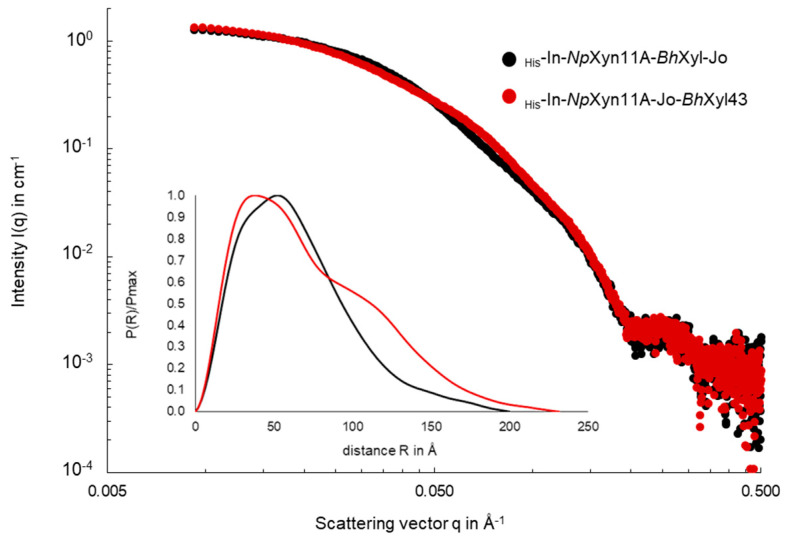
Superimposition of SAXS data from _His_-In-*Np*Xyn11A-*Bh*Xyl43-Jo and _His_-In-*Np*Xyn11A-Jo-*Bh*Xyl43 constructs (black and red dots respectively). The data are presented as a plot of Log I(q) in cm^−1^ vs. Log q in Å^−1^ to magnify the region of small angles. In the box below is represented the pair distribution function p(r) of each form, plotted as a function of P(r)/Pmax vs. r in Å.

**Figure 3 ijms-21-04360-f003:**
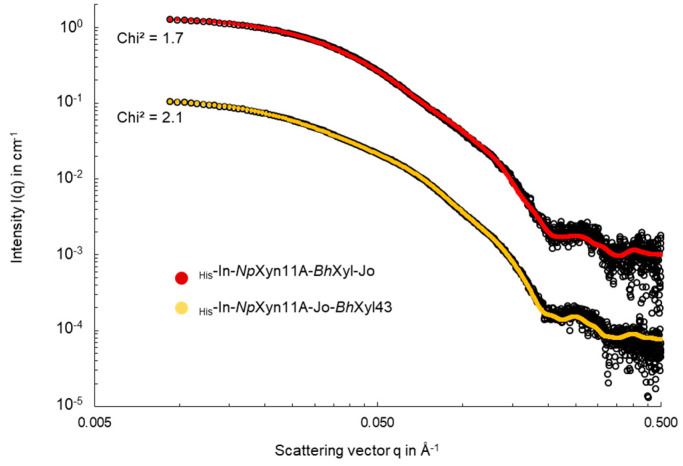
Superimposition of SAXS data from _His_-In-*Np*Xyn11A-*Bh*Xyl43-Jo and _His_-In-*Np*Xyn11A-Jo-*Bh*Xyl43 constructs and their structural models calculated with SASREFMX (red and orange line respectively). The data are presented as a plot of Log I(q) in cm^−1^ vs. Log q in Å^−1^ to magnify the region of small angles.

**Figure 4 ijms-21-04360-f004:**
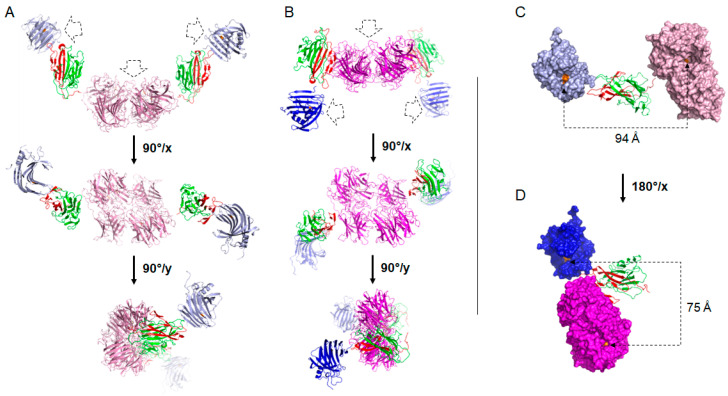
SAXS data compatible models of dimer of (**A**) _His_-In-*Np*Xyn11A-Jo-*Bh*Xyl43 arc shaped and (**B**) _His_-In-*Np*Xyn11A-*Bh*Xyl43-Jo “M” shaped. Structures are shown in three orientations. Arrow in drawn lines indicates access to catalytic pockets. (**C**) Details of monomeric organisation within the _His_-In-*Np*Xyn11A-Jo-*Bh*Xyl43 complex and (**D**) _His_-In-*Np*Xyn11A-*Bh*Xyl43-Jo complex. Drawn line indicates the distance separating the catalytic pocket of individual enzyme. Domain *Np*Xyn11A is in blue (light for _His_-In-*Np*Xyn11A-Jo-*Bh*Xyl43, dark for _His_-In-*Np*Xyn11A-*Bh*Xyl43-Jo), domain *Bh*Xyl43 is in magenta (light for _His_-In-*Np*Xyn11A-Jo-*Bh*Xyl43, dark for _His_-In-*Np*Xyn11A-*Bh*Xyl43-Jo), domain In is in green and domain Jo is in red. Catalytic residues are in orange.

**Figure 5 ijms-21-04360-f005:**
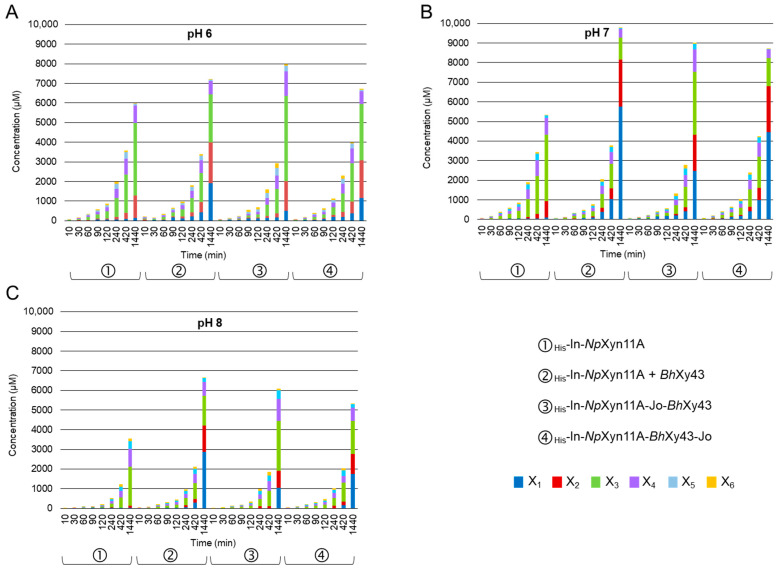
Evolution of the concentration of xylooligosaccharides from X_1_ to X_6_ at (**A**) pH 6, (**B**) pH 7 and (**C**) pH 8 over time determined by HPAEC-PAD. Hydrolysis conditions: 1 nM of enzymes, 37 °C 1% BWX during 24 h. Representation of the average of triplicate experiments (see Appendix A for details).

**Figure 6 ijms-21-04360-f006:**
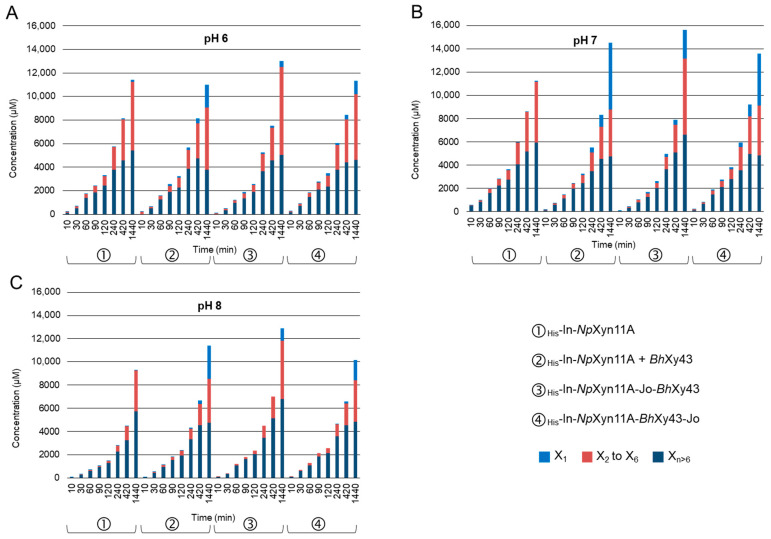
Evolution of the concentration of the total reducing equivalent xylose at (**A**) pH 6, (**B**) pH 7, and (**C**) pH 8 over time measured by DNS assay. Integrated with the DNS results, the concentration of xylose and xylooligosaccharides from DP 2 to 6, as determined by HPAEC-PAD, for the same samples. Hydrolysis conditions: 1 nM of enzymes, 37 °C 1% BWX during 24 h. Representation of the average of experiments in triplicate.

**Table 1 ijms-21-04360-t001:** Kinetic parameters of *Np*Xyn11A, *Bh*Xyl43 and the Jo In derivate enzymes on Beechwood xylan (BWX) and 4-nitrophenyl-β-d-xyloside (*p*NP-X).

Enzyme	BWX			*p*NP-X			References
	*K*_M app_ (mg·mL^−1^)	*k*_cat_ (10^3^ min^−1^)	*k*_cat_/*K*_M app_ (10^3^ min^−1^·mg^−1^·mL)	*K*_M_ (mM)	*k*_cat_ (s^−1^)	*k*_cat_/*K*_M_ (s^−1^·M^−1^)	
_His_-*Np*Xyn11A	0.75 ± 0.13	24.02 ± 1.51	32.02 ± 5.22	-	-	-	This work
_His_-In-*Np*Xyn11A	1.8 ± 0.7	46.1 ± 8.8	25.6 ± 6.5	-	-	-	[34]
_His_-*Bh*Xyl43	-	-	-	4.40 ± 0.50	12.09 ± 0.93	2750 ± 520	[37]
_His_-Jo-*Bh*Xyl43	-	-	-	6.01 ± 0.41	18.12 ± 1.17	3014.86 ± 10.88	This work

**Table 2 ijms-21-04360-t002:** Specific activities of *Np*Xyn11A and *Bh*Xyl43, as single enzyme or in complex. The 4-nitrophenyl-β-d-xyloside (*p*NP-X) substrate was used to measure xylosidase activity while 4-nitrophenyl-β-d-xylotrioside (*p*NP-X_3_) substrate was used to measure xylanase activity. Specific activity is defined as the µmoles of product formed per minute and per µmoles of enzyme. The value shown as ± mean standard deviation of replicate *n* = 3.

Single Enzymes	Chimeric Enzymes	Specific Activity (IU/µmole)	
		*p*NP-X	*p*NP-X_3_
_His_-*Bh*Xyl43		211.6 ± 5.3	-
_His_-Jo-*Bh*Xyl43		-	22.3 ± 11.4
_His_-In-*Np*Xyn11A		0.4 ± 0.5	1251.9 ± 15.2
*_His_*-In-*Np*Xyn11A *+*_His_-*Bh*Xyl43		222.9 ± 7.2	-
	*_His_*-In-*Np*Xyn11A-Jo-*Bh*Xyl43	118.0 ± 6.4	1404.7 ± 68.4
	*_His_*-In-*Np*Xyn11A-*Bh*Xyl43-Jo	177.3 ± 6.9	-

**Table 3 ijms-21-04360-t003:** Comparison of the activity of the different enzyme at various pH. Activity is expressed as the amount of µmoles of product formed per minute and per µmoles of enzymes. Hydrolysis conditions: 1 nM of enzymes, 37 °C, 1% BWX. UI = µmol·min^−1^. The value shown as ± mean standard deviation of replicate *n* = 3.

Enzyme	pH 6	pH 7	pH 8
	Activity 10^3^ (UI/µmole)	Activity 10^3^ (UI/µmole)	Activity 10^3^ (UI/µmole)
_His_-*Bh*Xyl43	0.36	0.36	0.24
_His_-In-*Np*Xyn11A	47.11 ± 1.29	37.35 ± 2.05	39.98 ± 1.73
_His_-In-*Np*Xyn11A + _His_-*Bh*Xyl43	41.76 ± 2.17	34.45 ± 0.71	36.03 ± 1.54
_His_-In-*Np*Xyn11A-Jo-*Bh*Xyl43	35.36 ± 1.95	31.02 ± 1.97	33.08 ± 1.07
_His_-In-*Np*Xyn11A-*Bh*Xyl43-Jo	41.81 ± 0.76	48.21 ± 1.16	36.79 ± 1.07

**Table 4 ijms-21-04360-t004:** Amount of equivalent of reducing sugar as determined by DNS assay after 24 h of hydrolysis at pH 7 and percentage of xylooligomers of DP ranging from 1 to 6 and oligomers of DP > 6. Hydrolysis conditions: 1 nM of enzymes, 37 °C 1% BWX after 24 h. Representation of the average of experiments in triplicate. The value shown as ± mean standard deviation of replicate *n* = 3.

Enzyme	Total Reducing Sugar (µM)	X_1_ (%)	X_2_ (%)	X_3_ (%)	X_4_ (%)	X_5_ (%)	X_6_ (%)	X_>6_ (%)
_His_-In-*Np*Xyn11A	11,278	0.8 ± 1.1	7.6 ± 0.5	29.8 ± 1.6	7.8 ± 0.6	1.1 ± 0.08	0.2 ± 0.07	52.6 ± 2.4
_His_-In-*Np*Xyn11A + _His_-*Bh*Xyl43	14,549	39.5 ± 1.5	16.4 ± 0.4	7.8 ± 0.2	2.9 ± 0.08	0.4 ± 0.01	0.1 ± 0.07	32.6 ± 1.6
*_His_*-In-*Np*Xyn11A-Jo-*Bh*Xyl43	15,640	15.8 ± 2.8	11.8 ± 0.7	20.4 ± 0.6	8.4 ± 0.2	1.9 ± 0.1	0.3 ± 0.1	42.3 ± 4.6
*_His_*-In-*Np*Xyn11A-*Bh*Xyl43-Jo	13,596	32.7 ± 1.7	17.3 ± 1.2	10.5 ± 0.6	3.2 ± 0.1	0.3 ± 0.1	0.2 ± 0.1	35.7 ± 3.2

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
