# Peer review of "Characterisation of the Effect of the Spatial Organisation of Hemicellulases on the Hydrolysis of Plant Biomass Polymer"

_ijms, 2020, doi:10.3390/ijms21124360_

Round 1
Reviewer 1 Report
The manuscript from Enjalbert T. et al reports a study on synergism related to bi-modular chimeric complexes connecting a xylanase and a xylosidase, both involved in the deconstruction of one of the major hemicellulose found in the plant cell wall. In this study the authors firstly described the different coupling engages and built via a recently described technics; the BioMolecular Welding toolbox. In addition, authors described the impact of different chimeras on the 3D structural shape by SAXS of the two catalytic modules and relate this to their respective biochemical (e.g. pH), but also catalytic parameters, i.e. specific activity and kinetic parameters. The stability of the different chimeras is also studied.
The work was well done, the methodology used is well suited to the proposed objectives, the results are fairly well discussed, the literature is reasonably covered, and the manuscript is clearly written. However, some minor points should be addressed, particularly with regard to optimal pH and the quality of some of the figures or even some of the data presented that should be added to the SI section to support the conclusions.
The authors should address the following points:
Page 1 - line 27: Authors wrote “…enzymes (15,640 and 14,549 μM of equivalent xylose, respectively).”, I suggested to add (i.e. 15,640 and 14,549 μM of equivalent xylose, respectively).
Page 2 - line 44: Add a comma between “[6], (ii)…”.
Page 2 - line 87: “that subtle” should be changed to “that subtile”.
Page 2/3 - line 93-95: It should be good to specify the residues from Jo and In fragments involved in the spontaneous isopeptide bridging, because at this stage it is not clear.
Page 3 - line 96: As the Figure 1B is firstly cited in the manuscript, the Figure 1B should be changed to Figure 1A and conversely Figure 1A to 1B.
Page 3 - line 105: Authors should add “to” in the sentence “This strategy was developed to allow…”.
Page 3 – line 114: I suggested to add the NMR data in the supplementary data files to support the conclusion of this sentence.
Page 3 - line 124: In the Figure 1A caption, the his-tag appearing in purple, it should be mentioned.
Page 4 – Line 135-142 and Page 8 – Line 290: Authors refers to Fig 1A/B SI and described twice times the pH optimum. I suggest combining both in a single paragraph.
Page 6 – line 224: Authors should refer to the Figure 4D.
Page 8 - line 259 and 260: I think it could be good to add the corresponding data in SI section to support both conclusions.
Page 8 – line 290: I am not sure that we can conclude to two pH optimum at 6 and 8 because i) we do not know if these results have been replicated and are significant (i.e. there is no s.d), ii) 6,5 and 7,5 might be necessary to conclude to a double peak, and iii) the figure 5 does not indicate an optimum at pH 8,0.
Page 9 – Line 307: Authors refers to the wrong figure and should consequently replace Fig 6 SI by Fig 4 SI.
Page 9 - line 332: Once again, I suggest that the authors add the corresponding data in SI, as they have the possibility to do so.
Page 10 – Figure 5: The figure 5 should be redesigned to be more readable.
Page 11 – Figure 6: The figure 6 should be redesigned to be more readable.
Page 12 – Line 434: Remove the space between “…strain…” and “…BL21…”
Reviewer 2 Report
This study explores the coordination of two enzymes that depolymerize xylan, a xylanase and a xylosidase. Broadly speaking the xylo-oligosaccharide product of xylanase is the substrate for xylosidase, resulting in the breakdown of the polymer into very small fragments. The authors have made constructs in which the two enzymes are fastened together in different relative orientations. The constructs are carefully characterized, then compared with a simple mixture of the two enzymes in assays of activity against chromogenic substrates and against beechwood xylan.
The construction of the complexes was achieved by the use of the recently developed ‘bio molecular welding’ technique, and this appears to have been successful in generating two distinct geometries for the orientation of the component enzymes, as judged from the low resolution structures that can be modelled on the basis of SAXS measurements. While the idea of artificially linking two enzymes is not new, the method of construction has not as far as I can see been applied to this problem before.
Measurements of activity are complicated by the different pH and temperature preferences of the two enzymes, but at pH 7 there is a striking difference between the specific molar activities of the two complexes with xylan as substrate (Table 3). The differences in total reducing sugar after 24h are less marked, but there are interesting variations in the production of oligosaccharides (Table 4).
Some minor but important issues:
Fig. 2 page 5: The caption indicates that there should be a box with a plot of pair distribution functions, but this is missing.
Table 2 page 8 and Table 3 page 9: Are the values shown as mean +/- SD? What is the number of replicates n?
Table 4 page 11: In this table we are told that n=3 but no error limits are shown. For this table in particular it is important to show the precision of the results so that the reader can assess the significance of the differences between the complexes.
Line 114 and lines 259, 260: both refer to unpublished data not shown in the paper. This may be a problem, depending on the policy of the journal.
